# Multimodal Learning and Reasoning for Visual Question Answering

**Ilija Ilievski**
Integrative Sciences and Engineering
National University of Singapore
ilija.ilievski@u.nus.edu

**Jiashi Feng**
Electrical and Computer Engineering
National University of Singapore
elefjia@nus.edu.sg

## Abstract

Reasoning about entities and their relationships from multimodal data is a key goal of Artificial General Intelligence. The visual question answering (VQA) problem is an excellent way to test such reasoning capabilities of an AI model and its multimodal representation learning. However, the current VQA models are over-simplified deep neural networks, comprised of a long short-term memory (LSTM) unit for question comprehension and a convolutional neural network (CNN) for learning single image representation. We argue that the single visual representation contains a limited and general information about the image contents and thus limits the model reasoning capabilities. In this work we introduce a modular neural network model that learns a multimodal and multifaceted representation of the image and the question. The proposed model learns to use the multimodal representation to reason about the image entities and achieves a new state-of-the-art performance on both VQA benchmark datasets, VQA v1.0 and v2.0, by a wide margin.

## 1   Introduction

One of the hallmarks of human intelligence is the ability to reason about entities and their relationships using data from different modalities [40]. Humans can easily reason about the entities in a complex scene by building multifaceted mental representations of the scene contents. Thus, any plausible attempt for Artificial General Intelligence must also be able to reason about entities and their relationships.

Deep learning models have demonstrated excellent performance on many computer vision tasks such as image classification, object recognition, and scene classification [37, 23]. However, the models are limited to a single task over single data modality and thus are still far from complete scene understanding and reasoning.

The visual question answering (VQA), a task to provide answers to natural language questions about the contents of an image, has been proposed to fill this gap [3, 22]. Solving the VQA task requires understanding of the image contents, the question words, and their relationships. The human-posed questions are arbitrary and thus besides being a challenging machine comprehension problem, their answering also involves many computer vision tasks such as scene classification, object detection and classification, and face analysis. Thus, VQA represents a problem comprised of multiple sub-problems over multimodal data and as such can serve as a proxy test for the general reasoning capabilities of an AI model [30].

However, the current state-of-the-art VQA approaches employ oversimplified deep neural network models, comprised of a long short-term memory (LSTM) unit [9] for question comprehension and a convolutional neural network (CNN) [24] for learning a single representation of the image. The single visual representation encodes a limited and general information about the image contents and

thus hampers the model's reasoning about the image entities and their relationships. The analysis of the model's behavior is also made difficult because of the single visual representation, as one does not have evidence of the visual information used by the model to produce the answer. The problem is partially alleviated by incorporating a visual attention module, as most VQA models do [6, 18, 45], but still, the VQA models remain black box in most cases. There have also been approaches that employ modular networks for VQA [2], however they achieved limited success. Most VQA models join the image and text representations via element-wise multiplication [18, 45], exception is the model proposed by Fukui et al. [6], which learns the joint representation via compact bilinear pooling. Goyal et al. [7] recently showed that the current state-of-the-art VQA methods actually learn dataset biases instead of reasoning about the image objects or learning multifaceted image representations.

In this work we introduce ReasonNet, a model that learns to reason about entities in an image by learning a multifaceted representation of the image contents. ReasonNet employs a novel multimodal representation learning and fusion module that enables the model to develop a complete image and question understanding. ReasonNet then learns to utilize the multiple image representations, each encoding different visual aspects of the image, by explicitly incorporating a neural reasoning module. In contrast to current VQA models, the ReasonNet model is fully interpretable and provides evidence of its reasoning capabilities. The proposed model outperforms the state-of-the-art by a significant margin on the two largest benchmark VQA datasets, VQA v1.0 [3] and VQA v2.0 [7]. In summary, the contributions of our work are as follows:

- We develop a novel multimodal representation learning and fusion method, crucial for obtaining the complete image understanding necessary for multimodal reasoning.
- We introduce a new VQA reasoning model that learns multifaceted image representations to reason about the image entities.
- We perform an extensive evaluation and achieve new state-of-the-art performance on the two VQA benchmark datasets.

## 2  Related work

**Multimodal reasoning models**  Recently, several works introduced modular neural networks for reasoning evaluated on natural language question answering and visual question answering [2, 1, 11]. The modular networks use conventional natural language parser [21] to obtain a network layout for composing the network architecture [11]. Later work also incorporated a dynamic network layout prediction by learning to rank the parser proposed modules [1]. The neural modules are then jointly trained to execute the task on which they are applied.

In contrast to our model, the existing modular networks learn to compose a network architecture using a hand-designed or parser proposed layout structure. These models were shown [2] to be unable to capture the complex nature of the natural language and perform poorly on the complex VQA v1.0 [3].

Concurrently to this work, Hu et al. [10] and Johnson et al. [16] proposed a similar modular network model that learns the network layout structure and thus do not require a parser. However, the models have been applied with success only to the synthetic dataset CLEVR [15] because for training the layout prediction module they require the ground-truth programs that generated the questions.

**Visual question answering**  The visual question answering (VQA) task has received great interest [5, 45, 50, 38] since the release of the first large-scale VQA v1.0 dataset by Antol et al. [3]. Typically, a VQA model is comprised of two modules for learning the question and the image representations, and a third module for fusing the representations into a single multimodal representation. The multimodal representation is then fed to multiple fully-connected layers and a softmax layer at the end outputs the probabilities of each answer (e.g. see [12, 6, 44]).

The question representation is learned by mapping each question word to a vector via a lookup table matrix, which is often initialized with word2vec [31] or skip-thought [20] vectors. The word vectors are then sequentially fed to a Long Short-Term Memory (LSTM) unit [9], and the final hidden LSTM state is considered as the question representation. The image representation is obtained from a pretrained convolutional neural network (CNN) [24], such as ResNet [8] or VGG [39], and the output from the penultimate layer is regarded as the image representation. Some, increase the information available in the image representation by using feature maps from a convolutional layer [45, 29, 6].

# 3 ReasonNet

We develop a novel modular deep neural network model that learns to reason about entities and their relationships in a complex scene using multifaceted scene representations. The model, named as ReasonNet, takes as inputs an image $I$ and a natural language text $L$. Then, ReasonNet passes the image and the text through its different modules that encode multiple image aspects into multiple vector representations. At the same time, ReasonNet uses a language encoder module to encode the text into a vector representation. Finally, ReasonNet's reasoning unit fuses the different representations into a single reasoning vector. In the following we give details of the network modules and its reasoning unit, while in Section 4 we ensemble and describe a ReasonNet applied to the VQA task (Figure 1).

## 3.1 Multimodal representation learning

ReasonNet incorporates two types of representation learning modules: visual classification modules and visual attention modules. A visual classification module outputs a vector that contains the class probabilities of the specific image aspect assigned to that module. While a visual attention module outputs a vector that contains visual features focused on the module's specific image aspect.

We denote the classification modules as $\phi$ and the visual attention modules as $\varphi$. We use bold lower case letters to represent vectors and bold upper case letters to represent tensors. Subscript indices denote countable variables while superscript indices label the variables. For example, $\boldsymbol{P}_k^\phi$ denotes the $k$-th matrix $P$ of the module $\phi$. In the following we describe each module type.

**Visual classification modules**   Each visual classification module maps a different aspect of the image contents to a module specific class representation space. Consequently, ReasonNet needs to transfer the output of each module to a common representation space. For example, if one module classifies the image scene as "train station", while another module classifies a detected object as "train", ReasonNet needs to be aware that the outputs are semantically related. For this reason, ReasonNet transfers each classification module's output to a common representation space using a lookup table matrix.

Formally, a classification module $\phi$ outputs a matrix $\boldsymbol{P}^\phi$ of $n$ one-hot vectors with lookup table indices of the $n$ highest probability class labels. The matrix $\boldsymbol{P}^\phi$ is then mapped to a vector $\boldsymbol{c}^\phi$ in a common representation space with:

$$\boldsymbol{c}^\phi = \mathrm{vec}(\boldsymbol{P}^\phi \boldsymbol{W}^{\mathrm{LT}}), \tag{1}$$

where $\boldsymbol{W}^{\mathrm{LT}}$ is the lookup table matrix with learned parameters.

**Visual attention modules**   ReasonNet passes the image $I$ through a residual neural network [8] to obtain a visual feature tensor $\boldsymbol{V} \in \mathbb{R}^{F \times W \times H}$ representing the global image contents. Then, each visual attention module $\varphi$ focuses the visual feature tensor to the specific image aspect assigned to the said module using an attention probability distribution $\boldsymbol{\alpha}^\varphi \in \mathbb{R}^{W \times H}$.

The module-specific visual representation vector $\boldsymbol{v}^\varphi \in \mathbb{R}^F$ is then obtained with:

$$\boldsymbol{v}^\varphi = \sum_{i=1}^{W} \sum_{j=1}^{H} \boldsymbol{\alpha}_{i,j}^\varphi \boldsymbol{V}_{i,j}. \tag{2}$$

**Encoder units**   A problem with this approach is that the common representation vectors $\boldsymbol{c}^\phi$ from the classification modules, being distributed representations of the modules' class labels, are high-dimensional vectors. On the other hand, the visual feature vectors $\boldsymbol{v}^\varphi$ of the attention modules are also high-dimensional vectors, but sparse and with different dimensionality and scale.

As a solution, ReasonNet appends to each classification and to each attention modules an encoder unit Enc that encodes a module's output vector $\boldsymbol{x}$ (equal to $\boldsymbol{c}^\phi$ if classification module and to $\boldsymbol{v}^\varphi$ if attention module) to a condensed vector $\boldsymbol{r}$ in a common low-dimensional representation space. The encoder units are implemented as two fully-connected layers followed by a non-linear activation function, and max pooling over the magnitude while preserving the sign:

$$\begin{aligned}
\boldsymbol{r} &= \mathrm{Enc}(\boldsymbol{x}) \\
\mathrm{Enc}(\boldsymbol{x}) &:= \mathrm{sgn}(f(\boldsymbol{x})) \cdot \max(|f(\boldsymbol{x})|), \\
f(\boldsymbol{x}) &:= \sigma\big(\boldsymbol{W}_2^E\big(\sigma(\boldsymbol{W}_1^E \boldsymbol{x} + \boldsymbol{b}_1^E)\big) + \boldsymbol{b}_2^E\big),
\end{aligned} \tag{3}$$

where $\boldsymbol{W}_k^E$ and $\boldsymbol{b}_k^E$ are the parameters and biases of the $k$-th layer in an encoder $E$, and $\sigma(\cdot) :=$ $\tanh(\cdot)$. The max pooling is performed over the row dimension as $f(\boldsymbol{x})$ outputs a matrix of row-stacked encoded vectors from one mini-batch.

**Text encoding**　ReasonNet treats the words in $L$ as class labels and correspondingly uses the lookup table from Eq. (1) and an encoder unit to map the text to a vector $\boldsymbol{r}^l$ in the same low-dimensional representation space of the other modules.

### 3.2 Multimodal reasoning

The reasoning capabilities of the ReasonNet model come from its ability to learn the interaction between each module's representation $\boldsymbol{r}_k$ and the question representation $\boldsymbol{r}^l$. The ReasonNet model parameterizes the $\boldsymbol{r}_k \times \boldsymbol{r}^l$ interaction with a bilinear model [42]. Bilinear models have excellent representational expressiveness by allowing the vector representations to adapt each other multiplicatively. The bilinear model is defined as:

$$\boldsymbol{s}_k = \boldsymbol{r}_k^\top \boldsymbol{W}_k^s \boldsymbol{r}^l + \boldsymbol{b}_k^s, \tag{4}$$

where $k = 1, \ldots, K$ and $K$ is the number of representation learning modules, provides a rich vector representation $\boldsymbol{s}^k$ of the $k$-th module's output $\times$ language interaction.

Note that while other multimodal representation learning works, e.g. [6], have criticized the use of bilinear models for representation fusion because of their high dimensionality as the tensor $\boldsymbol{W}_k^s$ is cubic in the dimensions of $\boldsymbol{r}_k$, $\boldsymbol{r}^l$, and $\boldsymbol{s}_k$. However, ReasonNet mitigates this issue by employing encoder units to reduce the dimension of the representation vectors and thus also reduce the dimension of the bilinear model. We evaluate different fusion methods with ablation studies in Section 5.3.

ReasonNet builds a complete image and language representation by concatenating each interaction vector $\boldsymbol{s}_k$ into a vector $\boldsymbol{g} = \|_{k=1}^K \boldsymbol{s}_k$, where $\|$ denotes concatenation of vectors. The vector concatenation is crucial for disentangling the contributions of each module in the model's task. By partially ignoring some of the inputs of the vector $\boldsymbol{g}$, ReasonNet can learn to "softly" utilize the different modules only when their outputs help in predicting the correct answer for the given question. In contrast to the recent works on module networks, e.g. [10, 16], ReasonNet can choose to partially use a module if it is helpful for the task instead of completely removing a module as other module network models do. For example, for the visual question answering (VQA) task, the soft module usage is particularly useful when answering a question which implicitly requires a module, e.g. answering the question "Is it raining?" implicitly requires a scene classification module.

The concatenation of the modules' representations also enables the interpretability of the reasoning behavior of our model. Specifically, by observing which elements of the vector $\boldsymbol{g}$ are most active we can infer which modules ReasonNet used in the reasoning process and thus explain the reasons for its behavior. We visualize the reasoning process on the VQA task in Section 5.4.

Finally, the multimodal representation vector $\boldsymbol{g}$ can be used as an input to an answer classification network, when applied to the VQA task, or as an input to an LSTM unit when applied to the image captioning task. In the next section, we use the challenging VQA task as a proxy test for ReasonNet reasoning capabilities.

## 4 ReasonNet for VQA

The visual question answering problem is an excellent way to test the reasoning capabilities of ReasonNet and its use of multifaceted image representations. For example, answering the question "Does the woman look happy?" requires a face detection, gender and emotion classification, while answering the question "How many mice are on the desk?" requires object detection and classification. Thus, for the VQA task, ReasonNet incorporates the following set of modules: 1. question-specific visual attention module, 2. object-specific visual attention module, 3. face-specific visual attention module, 4. object classification module, 5. scene classification module, 6. face analysis classification module. In the following we give a formal definition of the VQA problem and details of the model's modules when applied on the VQA task. The network architecture is visualized in Figure 1.

Namely, the VQA problem can be solved by modeling the likelihood probability distribution $p_{\text{vqa}}$ which for each answer $a$ in the answer set $\Omega$ outputs the probability of being the correct answer, given a question $Q$ about an image $I$:

$$\hat{a} = \arg\max_{a \in \Omega} p_{\text{vqa}}(a|Q, I; \boldsymbol{\theta}), \tag{5}$$

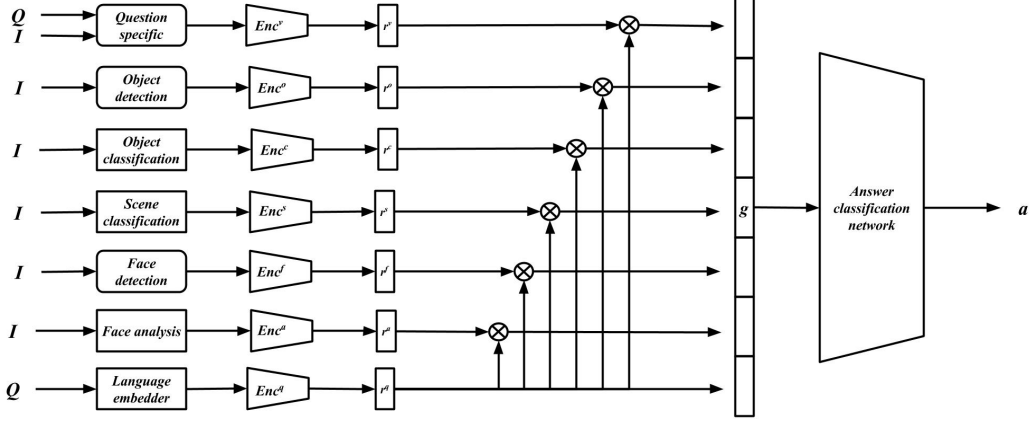

Figure 1: Network architecture diagram of the ReasonNet model applied on the VQA task. Round rectangles represent attention modules, squared rectangles represent classification modules, small trapezoids represent encoder units (Eq. (3)), thin rectangles represent the learned multimodal representation vectors, $\otimes$ represents the bilinear interaction model (Eq. (4)), and the big trapezoid is a multi-layer perceptron network that classifies the reasoning vector $\boldsymbol{g}$ to an answer $a$ (Eq. (7))

where $\boldsymbol{\theta}$ are the model parameters, $\hat{a}$ is the predicted answer, and $\Omega$ is the set of possible answers.

## 4.1 ReasonNet VQA modules

First, ReasonNet obtains the question representation vector $\boldsymbol{r}^l$ and a global visual feature tensor $\boldsymbol{V}$ as described in Section 3.1. ReasonNet then learns a question-specific image representation by using the question representation vector $\boldsymbol{r}^l$ to learn an attention probability distribution $\boldsymbol{\alpha}^v \in \mathbb{R}^{W \times H}$ over the global visual feature tensor $\boldsymbol{V} \in \mathbb{R}^{F \times W \times H}$ :

$$\boldsymbol{\alpha}^v = \mathrm{softmax}\left[\boldsymbol{W}^\alpha\Big(\sigma\big((\boldsymbol{W}^v\boldsymbol{r}^l + \boldsymbol{b}^v)\cdot \mathbb{1}\big) \circ \sigma(\boldsymbol{W}^V\boldsymbol{V} + \boldsymbol{b}^V)\Big) + \boldsymbol{b}^\alpha\right],$$

where $\sigma(\cdot) := \tanh(\cdot)$, $\boldsymbol{W}^\alpha, \boldsymbol{W}^v, \boldsymbol{W}^V$ and the corresponding biases are learned parameters, $\mathbb{1} \in \mathbb{1}^{W \times H}$ is used to tile the question vector representation to match the $\boldsymbol{V}$ tensor dimensionality, and $\circ$ denotes element-wise matrix multiplication. The question-specific visual feature vector $\boldsymbol{v}^v$ is then obtained with Eq. (2) and the representation vector $\boldsymbol{r}^v$ with Eq. (3).

Naturally, many of the VQA questions are about image objects so ReasonNet incorporates a fully convolutional network (FCN) [28] for object detection. Given an image, the FCN will output a set of object bounding boxes and their corresponding confidence scores. Each bounding box is represented as a four-element vector $\boldsymbol{d}^\top = [x, y, w, h]$, where $(x, y)$ is the coordinate of the top-left box corner and $w, h$ is the size of the box. Using a confidence score threshold ReasonNet obtains a set $\mathcal{B}$ containing high confidence bounding boxes. ReasonNet then uses the set $\mathcal{B}$ to compute an attention probability distribution $\boldsymbol{\alpha}^o$ that focuses the visual feature tensor $\boldsymbol{V}$ on the image objects.

To ground the image pixels to visual feature maps, all images are resized to the same pre-fixed dimension before feeding them to the object detection module. Thus each feature vector $\boldsymbol{v} \in \mathbb{R}^F$ from a corresponding element in the tensor slice $\boldsymbol{V}^{slice} \in \mathbb{R}^{W \times H}$ from the feature tensor $\boldsymbol{V} \in \mathbb{R}^{F \times W \times H}$ maps to a fixed sized image region.

Formally, for each $\boldsymbol{d}_k \in \mathcal{B}$, ReasonNet calculates a single object attention $\boldsymbol{\gamma}^k$. The $|\mathcal{B}|$ object-specific attentions are then used to calculate the overall object attention distribution $\boldsymbol{\alpha}^o$:

$$\boldsymbol{\alpha}^o = \mathrm{softmax}(\hat{\boldsymbol{\alpha}}^o),$$
$$\hat{\alpha}^o_{i,j} = \max_{k=1,\ldots,|\mathcal{B}|}(\gamma^k_{i,j}),$$
$$\gamma^k_{i,j} = \begin{cases} 1 & \begin{aligned} d^k_x &\leq i \leq (d^k_x + d^k_w), \\ d^k_y &\leq j \leq (d^k_y + d^k_h), \end{aligned} \\ 0.1 & \text{otherwise.} \end{cases} \tag{6}$$

As before, the object-specific visual feature vector $\boldsymbol{v}^o$ is then obtained with Eq. (2) and the representation vector $\boldsymbol{r}^o$ with Eq. (3).

ReasonNet further uses the object bounding boxes to obtain a vector of object class labels. Namely, for each bounding box in $\mathcal{B}$, ReasonNet crops out the image part corresponding to the box coordinates and then uses a residual network to classify the cropped-out image part and obtain a class label. The $n$ class labels of the boxes with highest class probability are represented as $n$ one-hot vectors of lookup table indices. The matrix $\boldsymbol{P}^c$, obtained by stacking the $n$ vectors, is then mapped to a dense low-dimensional vector $\boldsymbol{r}^c$ with Eq. (1) and Eq. (3).

Next, ReasonNet uses a scene classification network as many of the questions explicitly or implicitly necessitate the knowledge of the image setting. The scene classification network is implemented as a residual network trained on the scene classification task. As before, the top $n$ predicted class labels are represented as a matrix of $n$ one-hot vectors $\boldsymbol{P}^s$ from which the module's representation vector $\boldsymbol{r}^s$ is obtained (Eq. (1) and Eq. (3)).

Since the VQA datasets [7, 3] contain human-posed questions, many of the questions are about people. Thus, ReasonNet also incorporates a face detector module, and a face analysis classification module.

The face detector module is a fully convolutional network that outputs a set of face bounding boxes and confidence scores. As with the object detector, ReasonNet uses a threshold to filter out bounding boxes with low confidence scores and obtain a set of face detections $\mathcal{F}$. Then, from $\mathcal{F}$, using Eq. (6), ReasonNet obtains an attention probability distribution $\boldsymbol{\gamma}^f$ that focuses the visual feature tensor $V$ on people's faces. The face-specific visual feature vector $\boldsymbol{v}^f$ is then obtained with Eq. (2) and the representation vector $\boldsymbol{r}^f$ with Eq. (3).

The face bounding boxes from $\mathcal{F}$ are also used to crop out the image regions that contain a face and using a convolutional neural network to obtain three class labels for each detected face representing the age group, the gender, and the emotion. As with the other classification modules, ReasonNet represents the three class labels as a matrix of one-hot vectors $\boldsymbol{P}^a$ and uses Equations (1) and (3) to obtain the face analysis representation vector $\boldsymbol{r}^a$.

ReasonNet obtains a complete multimodal understanding by learning the interaction of the learned representations $\mathcal{H} = \{\boldsymbol{r}^v, \boldsymbol{r}^o, \boldsymbol{r}^c, \boldsymbol{r}^s, \boldsymbol{r}^f, \boldsymbol{r}^a\}$ with the question representation $\boldsymbol{r}^q$:

$$\boldsymbol{g} = \mathop{\big\|}_{\boldsymbol{r}_h \in \mathcal{H}} (\boldsymbol{r}_h^\top \boldsymbol{W}_h^s \boldsymbol{r}^q + b_h^s),$$

where $\boldsymbol{W}_h^s$ is a learned bilinear tensor for a representation $\boldsymbol{r}_h$, and $\|$ denotes concatenation of vectors.

Finally, ReasonNet forwards the vector $\boldsymbol{g}$, containing the question representation and multifaceted image representations, to a two-layer perceptron network which outputs the probabilities $p_\text{vqa}$(Eq. (5)):

$$p_\text{vqa}(a|Q, I; \boldsymbol{\theta}) = \text{softmax}\left[\sigma\big(\boldsymbol{W}_2^g \sigma(\boldsymbol{W}_1^g \boldsymbol{g} + \boldsymbol{b}_1^g) + \boldsymbol{b}_2^g\big)\right], \tag{7}$$

where $\boldsymbol{\theta}$ represents all the model parameters.

## 5 Experiments

### 5.1 Datasets

We evaluate our model on the two benchmark VQA datasets, VQA v1.0 [3] and VQA v2.0 [7]. The VQA v1.0 dataset is the first large-scale VQA dataset. The dataset contains three human-posed questions and answers for each one of the 204,721 images found in the MS-COCO [27] dataset. We also evaluate our model on the second version of this dataset, the VQA v2.0. The new version includes about twice as many question-answer pairs and addresses the dataset bias issues [7] of the VQA v1.0 dataset. We report results according to the evaluation metric provided by the authors of the VQA datasets, where an answer is counted as correct if at least three annotators gave that answer:

$$\text{Acc}(a) = \min(\frac{\sum_{j=1}^{10} \mathbb{1}(a = a_j)}{3}, 1).$$

For fair evaluation, we use the publicly available VQA evaluation servers to compute the overall and per question type results.

Table 1: Results of the ablation study on the VQA v2.0 validation.

| Method | All | Y/N | Num | Other | Q-type changed |
|---|---|---|---|---|---|
| VQA | 55.13 | 69.07 | 34.29 | 48.01 | |
| VQA+Sc | 56.80 | 70.62 | 35.14 | 49.99 | +2.74% Which |
| VQA+Sc+oDec | 58.46 | 71.05 | 36.16 | 52.86 | +5.73% What color is the |
| VQA+Sc+oDec+oCls | 59.82 | 72.88 | 37.38 | 54.47 | +3.68% How |
| VQA+Sc+oDec+oCls+fDec | 60.35 | **74.21** | **37.46** | 53.79 | +12.63% Is the man |
| VQA+Sc+oDec+oCls+fDec+fAna | **60.60** | 73.78 | 36.98 | **54.81** | +0.88% Is he |
| ReasonNet-HadamardProduct | 58.37 | 71.05 | 35.99 | 52.72 | |
| ReasonNet-MCB [6] | 58.78 | 71.04 | 36.96 | 53.35 | |
| ReasonNet | **60.60** | **73.78** | **36.98** | **54.81** | |

## 5.2 Implementation details

Given a question $Q$ about an image $I$, our model works as follows. Following [18, 6] the images are scaled and center-cropped to a dimensionality of $3 \times 448 \times 448$, then are fed through a ResNet-152 [8] pretrained on ImageNet [36]. We utilize the output of the last convolutional layer as the image representation $V \in \mathbb{R}^{2048 \times 14 \times 14}$.

The question words are converted to lowercase and all punctuation characters are removed. We further remove some uninformative words such as "a, "an", "the", etc. We then trim the questions to contain at most ten question words by removing the words after the first ten. The lookup table matrix uses 300-dimensional vectors, initialized with word2vec [31] vectors.

The module parameters used to produce the module's outputs are pretrained on each specific task and are kept fixed when applying ReasonNet to the VQA problem. In the following we give details of each module.

The object detection module is implemented and pretrained as in [32, 33]. The object classification and scene classification modules are implemented as ResNet-152, the only difference is the object classification module is pretrained on MS-COCO while the scene classification module is pretrained on Places365 [48, 49]. The object classification module outputs $80$ different class labels [27], while the scene classification module outputs $365$ class labels [48].

We implement and pretrain the face detection module following Zhang et al. [46, 47], while the age and gender classification is performed as [34, 35] and the emotion recognition following [25, 26]. The output of the age classification network is an integer from zero to hundred, so we group the integers into four named groups, $0 - 12$ as "child", $13 - 30$ as "young", $31 - 65$ as "adult" and $+65$ as "old". This enables as to map the integer outputs to a class labels. Similarly, the output from the gender classification module is $0$ for "woman" and $1$ for "man". Finally, the emotion recognition module classifies a detected face to the following seven emotions "Angry", "Disgust", "Fear", "Happy", "Neutral", "Sad", and "Surprise".

The encoder units encodes the module outputs to $500$-dimensional vectors, with a hidden layer of $1,500$ dimensions. Each bilinear interaction model outputs a $500$-dimensional interaction vector, i.e. $500 \times 500 \rightarrow 500$. The classification network classifies the reasoning vector $g$ using one hidden layer of $2,500$ dimensions to one of $4,096$ most common answers in the training set.

We jointly optimize the parameters of the encoder units, the bilinear models, and the answer classification network using Adam [19] with a learning rate of $0.0007$, without learning rate decay. We apply a gradient clipping threshold of $5$ and use dropout[41] (with $p(\text{keep}) = 0.5$) layers before and batch normalization[13] after each fully-conected layer as a regularization.

## 5.3 Ablation study

To assess the contribution of each ReasonNet module we perform an ablation study where we train a model that only uses one module and then subsequently add the rest of the proposed VQA modules. A VQA model with only question-specific attention module is denoted as "VQA", the addition of the scene classification module is denoted as "Sc", the object detection module as "oDec", the object classification module as "oCls", the face detection module as "fDec", and the face analysis as "fAna".

To evaluate the bilinear model as representation fusion mechanism, we compare ReasonNet to models where we only swap the bilinear interaction learning (Eq. (4)) with (1) Hadamard product (denoted as ReasonNet-HadamardProduct) and with (2) multimodal compact bilinear pooling [6] (denoted as ReasonNet-MCB). The bilinear interaction model maps the two vectors to an interaction vector, by learning a projection matrix $W$ that projects the vectorized outer product of the two vectors to an interaction vector. When using Hadamard product the interaction vector is just an element-wise multiplication of the two vectors. On the other hand, the MCB uses Count Sketch projection function [4] to project the two vectors to a lower dimension and then applies convolution of the two count sketch vectors to produce an interaction vector. As opposed to a bilinear model, the MCB does not learn a projection matrix. We train the models on the VQA v2.0 train set and evaluate them on the validation set. The results are shown in Table 1.

From Table 1 we can observe that each module addition improves the overall score. The results show that the object detection module is responsible for the highest increase in accuracy, specifically for question of type "Other". The addition of the object classification module further improves the accuracy on the "Other" question types, but the addition of the face detection module reduces the accuracy of "Other" question types and increases the accuracy on the "Yes/No" questions. Possible reasons for this is that the two attention modules (object detection and face detection) bring too much noise in the image representation. The increase in accuracy for the "Yes/No" questions is likely because most "Yes/No" questions are about people. Finally, the addition of the face analysis module brings the highest accuracy by returning the accuracy of the "Other" question types, possibly due to the face class labels help in understanding the face attention.

The results in Table 1 clearly show the representational expressiveness of the bilinear models as representation fusion. The bilinear model improves the accuracy for all question types, while there is a small difference between the Hadamard product and compact bilinear pooling, as discussed in [18].

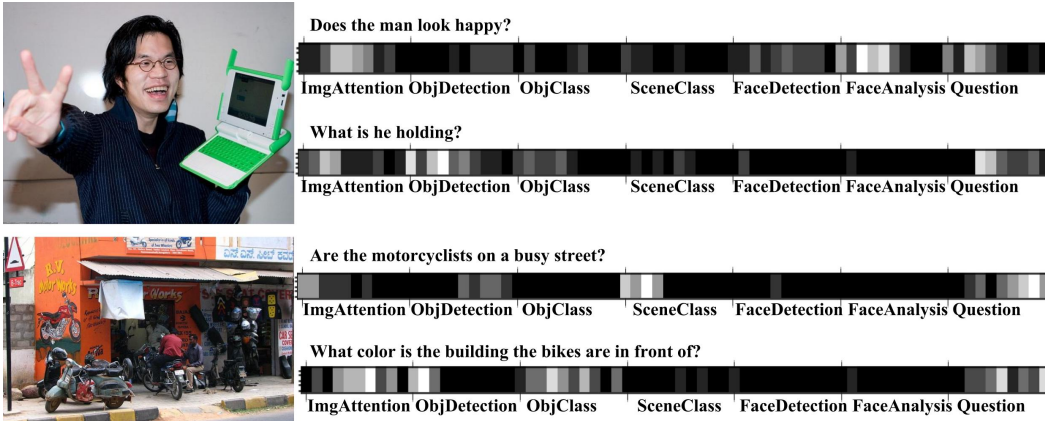

Figure 2: Qualitative results: we visualize the concatenation vector $g$ from Eq. (4.1) to investigate the module utilization given an image and two questions about the same image. The question-image pairs are from the VQA v2.0 test set.

## 5.4 Qualitative analysis and failure cases

To investigate the contribution of each of the network modules, we visualize the concatenation vector $g$ from Eq. (4.1) in Figure 2. We show two images from the VQA v2.0 test set and the corresponding $g$ vector values for two questions. Higher values are displayed with lighter shade of gray.

From Figure 2 we can observe that for the question "Does the man look happy?" the network strongly use the representation from the face analysis module and partially use the question-only representation and the question-specific attention representation. We can observe the same trend in the next two questions. It is interesting to observe that for complex questions such as "What color is the building the bikes are in front of?" most of the network modules are used which means the network does actually need multifaceted image representation to answer the questions.

The first example in Figure 2 also serves as a failure case. Namely, for the question "Does the man look happy?" the network correctly learns to use the face analysis module when the word "happy" is

Table 2: Results on the VQA v1.0 and v2.0 test-standard datasets for single models and without data augmentation. NMN is the only other modular neural. Results are taken from the official VQA evaluation servers.

| Method | VQA v1.0 test | | | | VQA v2.0 test | | | |
|---|---|---|---|---|---|---|---|---|
| | All | Y/N | Num | Other | All | Y/N | Num | Other |
| VQA-LangOnly | 48.9 | 78.1 | 34.9 | 27.0 | 44.26 | 67.01 | 31.55 | 27.37 |
| D-LSTM-nI [14] | 58.2 | 80.6 | 36.5 | 43.7 | 54.22 | 73.46 | 35.18 | 41.83 |
| NMN [2] | 58.7 | | | | | | | |
| DMN+ [43] | 60.4 | 80.4 | 36.8 | 48.3 | | | | |
| MRN [17] | 61.8 | 82.4 | 38.2 | 49.4 | | | | |
| HieCoAtt [29] | 62.1 | 79.9 | 38.2 | 51.9 | | | | |
| MCB [6] [1] | 64.7 | 82.5 | 37.6 | 55.6 | 62.27 | 78.82 | 38.28 | 53.36 |
| MLB [18] | 65.1 | 84.0 | 38.0 | 54.8 | | | | |
| **ReasonNet**[2] | **67.9** | **84.0** | **38.7** | **60.4** | **64.61** | **78.86** | **41.98** | **57.39** |

present in the question. However, the face analysis module incorrectly classifies the face as "Angry" misleading the network to give a wrong answer. Such error propagation from individual network modules is the main limitation of the proposed model. Future work can possibly overcome this limitation by backpropagating the error through the network modules. On the other hand, there is a constant improvement by the research community for each individual computer vision sub-task, that the limitation might be alleviated by simply incorporating the latest state-of-the-art network module.

## 5.5 Comparison with the state-of-the-art

Compared with the previous state-of-the-art on the VQA v1.0 dataset, the ReasonNet model achieves $2.8\%$ higher accuracy (Table 2). The improvement in accuracy predominately comes from the ability of the ReasonNet model to answer complex questions as evident from the $5.6\%$ increase in accuracy (denoted as "Other" in Table 2). This validates the ability of the ReasonNet to learn complex question-image relationships and to perform reasoning over the learned multimodal representations.

We observe the same improvement in accuracy of $2.34\%$ on the more challenging VQA v2.0 dataset. As on the VQA v1.0, the main improvement comes in answering the questions of type "Other" where there is a $4.03\%$ difference. The improvement in the "Other" questions likely comes from learning complex interactions of all modules outputs. There is also an improvement of $3.7\%$ in the counting questions denoted as "Num", which serves as evidence of the contribution of the object detection and object classification modules.

The new state-of-the-art on these datasets indicates the superiority of ReasonNet and the need for reasoning when answering complicated questions whose answering requires reasoning and understanding the relationship of multiple image objects.

## 6 Conclusion

We have presented a novel model for multimodal representation learning and reasoning. Our proposed reasoning model learns to reason over a learned multifaceted image representation conditioned on a text data. We validated the proposed reasoning neural network on the challenging VQA task and the model achieved a new state-of-the-art performance by a wide margin. The proposed reasoning model is general and probably applicable to other tasks involving multimodal representations, such as image captioning. We leave this promising direction for future work.

**Acknowledgments**

The work of Jiashi Feng was partially supported by National University of Singapore startup grant R-263-000-C08-133, Ministry of Education of Singapore AcRF Tier One grant R-263-000-C21-112 and NUS IDS grant R-263-000-C67-646.

## Footnotes

[1] Fukui et al. in [6] only report the *test-dev* results for VQA v1.0. The VQA v2.0 results are obtained from an implementation of their model.

[2] Due to a bug in the answer generating script the reviewed draft reported slightly lower VQA v2.0 results.

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
