[Reviews · NeurIPS 2017]

Reviewer 1



Summary -- The paper introduces a novel modular neural network for multimodal tasks such as Visual Question Answering. The paper argues that a single visual representation is not sufficient for VQA and using some task specific visual features such as scene classification or object detection would result in a better VQA model. Following this motivation, the paper proposes a VQA model with modules tailored for specific tasks -- scene classification, object detection/classification, face detection/analysis -- and pushes the state-of-the-art performance. Strengths -- -- Since VQA spans many lower level vision tasks such as object detection, scene classification, etc., it makes a lot of sense that the visual features tailored for these tasks should help for the task of VQA. According to my knowledge, this is the first paper which explicitly uses this information in building their model, and shows the importance of visual features from each task in their ablation studies. -- The formulation of their “module” structure is quite generic and many other modules corresponding to other low level computer vision tasks can be added to further improve the model performance. -- The approach has been explained in detail. Weaknesses -- -- The authors claim that their model is interpretable, but do not support this claim by any studies/analyses/qualitative examples. I agree that the ablation studies show the significance of each of the modules used in the proposed model, but it is not clear if the model is actually using the visual features from the correct module for a given question, e.g. for the question “Does the woman look happy?”, is it really the face analysis module which provides the correct signal to accurately predict the answer? This can be studied by analysing (e.g. making a pie chart of) the questions whose predictions were incorrect when the face analysis module was absent but are correct after adding the face analysis module. Similar studies can be done for all the modules. Without any such analysis, I am not convinced if the model is actually doing what the motivation of the paper says. If the author can report any studies/analyses supporting this, I am happy to increase my rating. -- The paper does not talk about the weaknesses of the proposed model. Discussions, ideally using failure cases, about limitations of the proposed model and what is needed to improve it are very important for continuous research in this area and should be an integral part of any paper. -- I am confused about the mathematics of some equations. In Eq 1, how is the matrix multiplication of 2 matrices P and W resulting in a vector c? In Eq 3, what dimension is the max pooling happening over? In Eq 4, should it be transpose of r_{k} which is being premultiplied with tensor W_{k}? -- In Table 2, the MCB numbers reported are test-dev numbers instead of test-standard. MCB only reports test-standard numbers for their ensemble model with data augmentation. Please fix. -- For classification modules, have the authors tried using soft probability scores instead of picking top k class labels? -- Minor: - A lot of cited papers have been published. It might be better to cite the conference version instead of arXiv. - r_{q} and r^{l} has been used interchangeably. At many places, subscripts and superscripts have been used interchangeably. Please fix. - Line 9: “does not limit” → “limits” After author rebuttal: I thank the authors for the additional analyses. The rebuttal addresses all my concerns. I think this paper introduces an interesting and novel approach for VQA, which would be useful to the VQA research community. Therefore, I recommend the paper for acceptance. I have changed my rating to 7.

Reviewer 2



Summary This paper proposes an approach that combines the output of different vision systems in a simple and modular manner for the task of visual question answering. The high level idea of the model is as follows. The question is first encoded into a bag of words representation (and passed through an MLP). Then different vision systems (which extract raw features, or compute attention on images using object detection outputs or face detection outputs or scene classification outputs) are all condensed into a representation compatible with the question. Finally the approach takes an outer product between the image representations from various tasks and the question, and concatenates the obtained representations. This concatenated feature is fed as input to an answering model which produces distributions over answer tokens. The entire model is trained with max-likelihood. Results on the VQA 1.0 as well as VQA 2.0 datasets show competitive performance with respect to the state of the art. Strengths 1. At a high level the proposed approach is very well motivated since the vqa task can be thought of as an ensemble of different tasks at different levels of granularity in terms of visual reasoning. The approach has flavors of solving each task separately and then putting everything together for the vqa task. 2. The results feature ablations of the proposed approach which helps us understand the contributions of different modules in achieving the performance gains. 3. It is really encouraging that the approach obtains state of the art results on VQA. Traditionally there has been a gap between modular architectures which we “feel” should be good / right for the task of VQA and the actual performance realized by such models. This paper is a really nice contribution towards integrating different vision sub-problems for VQA, and as such is a really good step. Weakness 1. When discussing related work it is crucial to mention related work on modular networks for VQA such as [A], otherwise the introduction right now seems to paint a picture that no one does modular architectures for VQA. 2. Given that the paper uses a billinear layer to combine representations, it should mention in related work the rich line of work in VQA, starting with [B] which uses billinear pooling for learning joint question image representations. Right now the manner in which things are presented a novice reader might think this is the first application of billinear operations for question answering (based on reading till the related work section). Billinear pooling is compared to later. 3. L151: Would be interesting to have some sort of a group norm in the final part of the model (g, Fig. 1) to encourage disentanglement further. 4. It is very interesting that the approach does not use an LSTM to encode the question. This is similar to the work on a simple baseline for VQA [C] which also uses a bag of words representation. 5. (*) Sec. 4.2 it is not clear how the question is being used to learn an attention on the image feature since the description under Sec. 4.2 does not match with the equation in the section. Speficially the equation does not have any term for r^q which is the question representation. Would be good to clarify. Also it is not clear what \sigma means in the equation. Does it mean the sigmoid activation? If so, multiplying two sigmoid activations (with the \alpha_v computation seems to do) might be ill conditioned and numerically unstable. 6. (*) Is the object detection based attention being performed on the image or on some convolutional feature map V \in R^{FxWxH}? Would be good to clarify. Is some sort of rescaling done based on the receptive field to figure out which image regions belong correspond to which spatial locations in the feature map? 7. (*) L254: Trimming the questions after the first 10 seems like an odd design choice, especially since the question model is just a bag of words (so it is not expensive to encode longer sequences). 8. L290: it would be good to clarify how the implemented billinear layer is different from other approaches which do billinear pooling. Is the major difference the dimensionality of embeddings? How is the billinear layer swapped out with the hadarmard product and MCB approaches? Is the compression of the representations using Equation. (3) still done in this case? Minor Points: - L122: Assuming that we are multiplying in equation (1) by a dense projection matrix, it is unclear how the resulting matrix is expected to be sparse (aren’t we mutliplying by a nicely-conditioned matrix to make sure everything is dense?). - Likewise, unclear why the attended image should be sparse. I can see this would happen if we did attention after the ReLU but if sparsity is an issue why not do it after the ReLU? Perliminary Evaluation The paper is a really nice contribution towards leveraging traditional vision tasks for visual question answering. Major points and clarifications for the rebuttal are marked with a (*). [A] Andreas, Jacob, Marcus Rohrbach, Trevor Darrell, and Dan Klein. 2015. “Neural Module Networks.” arXiv [cs.CV]. arXiv. http://arxiv.org/abs/1511.02799. [B] Fukui, Akira, Dong Huk Park, Daylen Yang, Anna Rohrbach, Trevor Darrell, and Marcus Rohrbach. 2016. “Multimodal Compact Bilinear Pooling for Visual Question Answering and Visual Grounding.” arXiv [cs.CV]. arXiv. http://arxiv.org/abs/1606.01847. [C] Zhou, Bolei, Yuandong Tian, Sainbayar Sukhbaatar, Arthur Szlam, and Rob Fergus. 2015. “Simple Baseline for Visual Question Answering.” arXiv [cs.CV]. arXiv. http://arxiv.org/abs/1512.02167.